# Polymer Foam Concrete FC500 Material Behavior and Its Interaction in a Composite Structure with Standard Cement Concrete Using Small Scale Tests

**DOI:** 10.3390/polym14183786

**Published:** 2022-09-10

**Authors:** Daniel Papán, Daniel Ďugel, Zuzana Papánová, Martin Ščotka

**Affiliations:** Faculty of Civil Engineering, Department of Structural Mechanics and Applied Mathematics, University of Žilina, Univerzitná 8215/1, 010 26 Žilina, Slovakia

**Keywords:** polymer foam concrete, cement concrete, experimental measurement, stress–strain diagram, numerical model, analytical model

## Abstract

This paper focuses on the investigation of the material properties of FC500 foam concrete. Innovation is very important for the solution of cast-in-place concrete forms in practice today. Part of its innovative construction application is the possibility of using foam concrete in a composite structure and the use of its mechanical properties in the load-bearing parts of civil engineering structures. The method of detecting the mechanical properties of foam concrete by using non-standard cantilever test is also innovative. Here, an advanced approach of modelling specimens using powerful computational systems based on the finite element method is used. This modern material is researched especially for its use in transportation structures. For its application, it is necessary to define its resistance to mechanical loads. The main content of the research consists of correlations between experimental measurements and analytical and numerical results. This is the principle of quasi-linear identification of the non-linear behavior of polymeric cementitious porous material during tests on specimens. The focus of the research is an extensive experiment: measurements of the deformation of the specimens until failure. The following methods were chosen to investigate the material properties: small cantilever test, standard tensile test and compression test. The cantilever test was performed for the individual components of the FC500 composite and cement concrete, but also as a compact composite. Numerical simulation models were developed to correlate the individual results in order to validate the uniaxial test results. The conclusions of the research led to the definition of standardized stress–strain diagrams of the FC500 material for compression and especially tension. This is a definition of the behavior of this polymer composite, usable for the development of numerical models of full-scale structures. The results of the research will be applied in the development of national standards for the use of advanced materials in transportation structures (cycle paths, parking lots, traffic playgrounds, lightly trafficked forest roads and trails, etc.).

## 1. Introduction

The first foam concrete (FC) appeared with the Romans, who discovered that adding animal blood to a mixture of small stones, sand and hot lime made the mixture easier to work with, more durable and more porous. Nevertheless, the first concrete composed of a mixture of Portland cement and foam was not patented until 1923 by Axel Eriksson. An initial comprehensive survey was further carried out by Valore (1954) [1] on aerated concrete [2] in 1954, followed by a detailed investigation by Rudnai, Shorth and Kinniburgh in 1963 [3]. Over the last 25 years, the foaming agents used for the preparation of foam concrete, as well as its method of preparation and its admixtures, have improved considerably [4]. Compared to ordinary concrete, foam concrete has many advantages.

### 1.1. Properties of Foam Concrete

It is a porous material that is created by mixing technical foam, cement concrete (CC) and admixtures in varying proportions to achieve different bulk weights ranging between 100 kg/m^3^ and 1600 kg/m^3^ [1,5,6]

The material has many advantages that contribute to its potential for use in the construction industry, used as a filling material, thermal insulation or as material helping to maintain suitable indoor conditions.

Due to its dense cell structure, the material is compressed on impact, thus increasing its ability to absorb kinetic energy.

As foam concrete consists of a solid matrix structure, the material is not very sensitive to seismic shock waves, so it is also suitable for use on liquefaction-prone soils. (Recommended by the U.S. Bureau of Reclamation in fault zones) [5].

Typical densities of cast-in-place foam concrete range between 100 and 1600 kg/m^3^, with 28-day and its compressive strengths from 0.1 up to 18.0 N/mm^2^. Due to its low density, foam concrete causes low vertical loads on foundation structures. This is a particularly important property in areas with poor foundation conditions [6,7].

After hydration, foam concrete forms a strong, well bonded structure. It is an efficient and free-standing (monolithic) structure and does not cause lateral loads on adjacent structures after curing [8].

Foam concrete allows construction cost savings as it can be applied directly to existing foundation conditions such as peat or poor soils. Its self-supporting nature allows reduced construction costs for earth retaining structures. Foam concrete allows construction on unconformable ground and reduces the need for pile foundations. High-volume equipment with quick installation reduces fabrication costs. Foam concrete requires lower maintenance costs due to the durability of foam concrete and low slump [5,9,10].

Foam concrete blocks are lightweight, large and precise in size. This increases the speed of construction several times. Additionally, it is possible and easier to create openings for the technical equipment of the building [5,10].

Due to its low weight, this material can be used to replace backfill soil and some underground structures [5], for example, around underground garages or large-span underground window wells, in addition to underground communications in case of soil falling out of the vault during construction.

### 1.2. Testing Methods of Foam Concrete

Many research teams around the world are investigating the properties of foam concrete in general. Laboratory tests on foamed concrete specimens are based on the testing methodology of standard cement concrete. The individual shapes used in the tests are also the subject of research. However, the most commonly used specimen for tensile tests is the dog-bone (piscot) shape. As in other tensile tests of polymeric materials. The test of material properties in compression is usually realized on a cube or cylinder-the specimen. However, the most important test is the bending test, which is performed on beams of cuboid shape. In terms of applicability, most research has focused on the four-point and, less rarely, the three-point bending test. This is of course due to the most common loading when using cement concrete and foam concrete.

Worldwide research on foam concrete has been summarized by authors from China under the title “Foam Concrete: A State-of-the-Art and State-of-the-Practice Review” [11]. This is a comprehensive document summarizing the knowledge on foam concrete production, processing, design, testing and use. Similar topics to those presented in Chapters 2 and 3 have been addressed by Australian and Chinese researchers, focusing on the compressive properties of foam concrete [12]. The analysis of tensile testing and its possible application to cementitious materials has been addressed by Chinese researchers in the literature review [13] again. However, most scientific teams are focused on the testing of material properties of foam concrete in bending: the work of Polish experts is associated with this problem in particular [14]. Regarding domestic research in Slovakia, the most devoted research focusing on the presented problems are by M. Decký, M. Drusa and W. Scherfel [15,16]. 

## 2. Methodology

### 2.1. Input Parameters

Linearized mathematical models can be used for part of the foam concrete redesign. They inaccurately determine the assumed theoretical beam deflection.

From several investigations, it is possible to take the basic material characteristics of a foam concrete of the chosen quality P500, whose bulk density is *ρ* ≈ 500 kg/m^3^.

Previous work from the University of Žilina have already made some assumptions, and it is possible to take some of these parameters from them. They are listed in Table 1 as elastic modulus “*E*”, Poisson’s constant “*v*”, maximum compressive strength “*f_c_*”, maximum tensile strength “*f_t_*” and maximum flexural strength “*σ*_0_”. Some listed parameters were taken from other foreign authors. [8,17,18,19,20,21].

It can be seen from the review that the parameter values are inconsistent, which may be due to the variation in the size and number of pores in the foam concrete when testing the specimens. The selected material characteristics will be further used in a linear calculation using the double integration method and will be compared with the measured results of the experiment itself. They will also be compared with the numerical model using the finite element method (“FEM”) in ANSYS. For the calculation of the linear model, known linear material characteristics will be used

### 2.2. Double Integration Method–Analytical Investigation Used for the Foam Concrete Models

The double integration method is based on the fundamentals of the mechanics and geometry of the bending beam (Figure 1). A moment-loaded beam can be defined as being bent or stressed into the shape of a circle after extension.

From the principles of Hooke’s law, we know the expression for the proportional transformation
(1)$$\varepsilon =\frac{∆L}{L_{0}}=\frac{CD-AB}{AB}=\frac{\left(R+y\right)*\theta -R*\theta }{R*\theta }$$

Simplifying Equation (1), we express relative transformation as a function of radius and distance from the neutral axis
(2)$$\varepsilon =\frac{y}{R}$$

Since considered a positive distance *y* and it points downwards, the ratio transformation is positive.

Using Hooke’s law in its standard form, we can derive the bending stresses on the beam
(3)$$\sigma =E*\varepsilon =\frac{E*y}{R}$$

What is essential in the calculation is the relationship of the bending moment and the stresses in the cross-section. The resulting stresses represent a standard stress waveform across the triangular cross-section with increasing stress from the neutral axis to the edges of the beam. Based on their area, we can define the bending moment acting in an exact point of the cross-section as
(4)$$M=\frac{E}{R}\int_{A}^{\;}y^{2}dA$$

From the result, part of the surface integral equation is the definition of the quadratic moment of the cross-section area, referred to as “*I*”, which defines the resistance of the cross-section to bending moment. Additionally, “*W*” is cross-section parameter–modulus depending on the height of the cross-section. By rearranging the equation, it is possible to obtain
(5)$$M=\frac{EI}{R}$$

By substituting into Equation (3), we obtain the bending equation
(6)$$\sigma =\frac{M}{I}*y=\frac{M}{W}$$

From Equation (5), it is further possible to derive the moment curvature equation
(7)$$\frac{1}{R}=\frac{M}{EI}$$

There is an equation that can accurately determine the radius of a curve at any point of its path [22]
(8)$$\frac{1}{R}=\frac{\frac{d^{2}y}{dx^{2}}}{{\left[1+{\left(\frac{dy}{dx}\right)}^{2}\right]}^{3/2}}=\frac{M}{EI}$$

However, this equation is quite complicated, and can be simplified since the elastic state of the deformed body is assumed. We can say that the slope *dy*/*dx* approaches zero and since it is squared it approaches zero even more. Then, it is possible to consider the value below the fractional line as 1 and thus the equation is modified to a simplified form:(9)$$\frac{1}{R}=\frac{M}{EI}=\frac{d^{2}y}{dx^{2}}$$

This equation expresses the deformation of the beam if we consider the Navier Bernoulli hypothesis and the elastic part of Hooke’s law. However, to express the deflection accurately, it is necessary to derive the deformation value “*y*”. Hence, double integration term comes to apply
(10)$$y=\int \int \frac{M}{EI}dx\;dx$$

In the case of the static cantilever model that was used in the experiment, the following derived equations can be used to calculate the bending moment on the cantilever as shown in Figure 2.

Thus, a schematic of the actual structure was constructed for the experiment, where the distance of the imposed force *F* was *L* = 260 mm and the force increased with time. The deflection of the specimen was measured at two locations, namely *L′* = 220 mm, which was subsequently evaluated and at *L″* = 160 mm. 

Therefore, it is possible, on the basis of the bending moment, to derive the deflection by the double integration method as follows
(11)$$\int \int \frac{d^{2}y}{dx^{2}}dx\;dx=\frac{M\left(x\right)}{EI}=\int \int \frac{-P*x}{EI}dx\;dx$$
(12)$$\int \frac{dy}{dx}\;dx=\frac{1}{EI}*\int \left(-\frac{P}{2}x^{2}+C_{1}\right)dx$$
(13)$$y=\frac{1}{EI}*\left(-\frac{P}{6}x^{3}+C_{1}x+C_{2}\right)$$

By double integrating the original equation, it derives the equation of the deflections. The only problem remaining are the integration constants, which are unknown. However, these can be derived from the boundary conditions of the system. The interference of the beam on the first side gives the conditions that the cross-section cannot rotate or shift, which can be expressed mathematically as
(14)$$x=L\to \frac{dy}{dx}=0$$
(15)$$x=L\to \frac{d^{2}y}{dx^{2}}=0$$
by substituting into the equations, the integration constants *C*_1_ and *C*_2_ can be expressed.
(16)$$C_{1}=-\frac{P}{2}L^{2},$$
(17)$$C_{2}=-\frac{P}{3}L^{3}$$

The result is an equation of beam deflection that can be applied to evaluate the linear deformations of foam concrete, using the material characteristics of the modulus of elasticity and the dimensions of the specimen.
(18)$$y\left(x\right)=\frac{-F*L{\left(x\right)}^{3}}{3EI}$$

However, this method does not appear to be very accurate as it is based on the Navier Bernoulli hypothesis and also assumes that the elastic modulus of the material remains constant throughout the loading period until failure. However, this has not been shown in experimental measurements.

### 2.3. Experimental Measurements

For the experimental measurements, a measuring setup was used, which is assembled in the structural mechanics laboratory of the university. It was adapted to perform the cantilever test. Previous tensile testing of the specimens was incorporated for the design and the experience for this measurement was used. The foam concrete specimens themselves were provided by the company manufacturing the material. They are manufactured using a precision process and called FC500. They were experimentally tested by cantilever testing and based on the results. The stress–strain diagram of the material was evaluated using ANSYS workbench 2022 R1 software.

The measurement mechanism used to design the experiment is simply summarized in scheme of experimental measuring setup, which is seen in Figure 3, Figure 4 and Figure 5. It is made of a solid steel block supporting the measured specimen. The loading force is applied through a steady automatic electric system. The calibration of applied force was created by Dr. Ing. George Wazau. Deformations are measured as a pair of indicator deflectometers, and their measurement values are written instantly onto a data disk.

### 2.4. Finite Element Method–Numerical Method Used for the Foam Concrete Models

Today, the most widely used numerical method for solving mechanics problems is the Finite element method. However, it is a numerical method with a large number of equations and it is inherently used with modern computing technology.

The construction and calculation procedure in the finite element method is divided into phases:Discretization of the body to a finite number of elementsApproximation of force or deformation quantities on each single elementIntegration of finite elements into a whole while maintaining continuity of deformationsEnergy minimization-solving boundary condition equations and determining unknown nodal parametersDetermination of unknowns at each finite element, and hence calculation of internal forces [23].

#### Principle of the Finite Element Method

The essence of the method is to divide the structure under analysis into a system of finite elements which are connected to each other at nodes. For such a system to function, it is necessary that it satisfies the conditions of continuity and equilibrium at these nodes.

The theory assumes that if the elements are one-dimensional (members, beams), two-dimensional (plates, walls, shells) or three-dimensional (solids), their vectors of displacement components of points are related to each other. Thus, a change in the position of a point is said to be expressed by a displacement vector field
(19)$$\left\{u\right\}=\left\{\begin{array}{c} u\left(x,y,z\right)\\ v\left(x,y,z\right)\\ \varphi \left(x,y,z\right) \end{array}\right\}$$

In this case, the strain field tensors must also follow each other
(20)$$\left\{\varepsilon \right\}={\left\{\varepsilon_{x},\varepsilon_{y},\varepsilon_{z},\gamma_{xy},\gamma_{yz},\gamma_{xz}\right\}}^{T}$$

Additionally, last but not least, the tensor stress field
(21)$$\left\{\sigma \right\}={\left\{\sigma_{x},\sigma_{y},\sigma_{z},\tau_{xy},\tau_{yz},\tau_{xz}\right\}}^{T}$$

Since this is the result of the calculation and therefore the finding of the displacements of the nodes, the deformation of the nodes and the stresses in the nodes, it is necessary to reach them in some way. It is possible to use Cauchy’s system of fifteen equations for the computation, which can be solved by considering explicit geometric and force boundary conditions as follows:

Differential equations of equilibrium
(22)$$\left[\partial \right]\left\{\sigma \right\}-\left\{b\right\}=\left\{0\right\}\;na\;V$$

Geometric equations
(23)$${\left[\partial \right]}^{T}\left\{u\right\}-\left\{\varepsilon \right\}=\left\{0\right\}\;na\;V$$

Physical equations
(24)$$\left\{\varepsilon_{0}\right\}+\left[C\right]\left\{\sigma \right\}=\left\{\varepsilon \right\}$$
(25)$$\left[D\right]\left(\left\{\varepsilon \right\}-\left\{\varepsilon_{0}\right\}\right)=\left\{\sigma \right\}$$

Static boundary equations
(26)$$\left[n\right]\left\{\sigma \right\}-{\left\{p\right\}}_{n}=\left\{0\right\}\;\mathrm{na}\;\mathrm{Sp}$$
(27)$$\left[n\right]\left\{\sigma \right\}-{\left\{\sigma \right\}}_{n}^{s}=\left\{0\right\}\;\mathrm{na}\;\mathrm{Ss}+\mathrm{Su}$$

Kinematic boundary conditions
(28)$$\left[n\right]\left\{u\right\}-{\left\{\bar{u}\right\}}_{n}^{s}=\left\{0\right\}\;\mathrm{na}\;\mathrm{Su}$$
(29)$$\left[n\right]\left\{u\right\}-{\left\{\bar{u}\right\}}_{n}^{p}=\left\{0\right\}\;\mathrm{na}\;\mathrm{Ss}+\mathrm{Sp}$$
where the matrix [*C*] is the material’s compliance matrix [18]:


(30)
$$\left[C\right]=\left[\begin{array}{cccccc} 1/E_{x} & -v_{xy}/E_{x} & -v_{xz}/E_{x} & 0 & 0 & 0\\ -v_{yx}/E_{y} & 1/E_{y} & -v_{yz}/E_{y} & 0 & 0 & 0\\ -v_{zx}/E_{z} & -v_{zy}/E_{z} & 1/E_{z} & 0 & 0 & 0\\ 0 & 0 & 0 & 1/G_{x} & 0 & 0\\ 0 & 0 & 0 & 0 & 1/y & 0\\ 0 & 0 & 0 & 0 & 0 & 1/G_{z} \end{array}\right]$$


The matrix is defined by nine constants, since the elements of the first submatrix are bound by the symmetry conditions *v_xy_/E_y_ = v_yz_/E_z_; v_yz_E_z_ = v_zy_/E_y_; v_zx_/E_x_ = v_xz_/E_z_*.

The inverse matrix to the compliance matrix is the material stiffness matrix [*D*] and its form is
(31)$${\left[C\right]}^{-1}=\left[D\right]=\left[\begin{array}{cccccc} d_{xx} & d_{xy} & d_{xz} & 0 & 0 & 0\\ d_{yx} & d_{yy} & d_{yz} & 0 & 0 & 0\\ d_{zx} & d_{zy} & d_{zz} & 0 & 0 & 0\\ 0 & 0 & 0 & G_{xy} & 0 & 0\\ 0 & 0 & 0 & 0 & G_{yz} & 0\\ 0 & 0 & 0 & 0 & 0 & G_{zx} \end{array}\right]$$

The stiffness matrix is based on the basic Hooke’s law as a one-element matrix of the form
(32)$$\left\{\sigma \right\}=\left[D\right]\left\{\varepsilon \right\},\;\;\;\;and\;\left[D\right]=E$$

The vector {*b*} is the vector of volumetric forces on the element and the vector {*p*} is the vector of surface forces
(33)$$\left\{b\right\}={\left\{b_{x},b_{y},b_{z}\right\}}^{T}$$
(34)$$\left\{p\right\}={\left\{p_{x},p_{y},p_{z}\right\}}^{T}$$

From the bulk forces and stress balance conditions on the differential element of a solid, it is possible to obtain the operator matrix [*∂*] used in the relations of the geometric and physical equations
(35)$$\left[\partial \right]=\left[\begin{array}{cccccc} \frac{\partial }{\partial x} & 0 & 0 & \frac{\partial }{\partial y} & 0 & \frac{\partial }{\partial z}\\ 0 & \frac{\partial }{\partial y} & 0 & \frac{\partial }{\partial x} & \frac{\partial }{\partial z} & 0\\ 0 & 0 & \frac{\partial }{\partial z} & 0 & \frac{\partial }{\partial y} & \frac{\partial }{\partial x} \end{array}\right]$$
cosines [*n*] of the external normal *n_x_, n_y_, n_z_* to the surface *S* in the relations of static boundary conditions and kinematic boundary conditions have an arrangement similar to the operator matrix
(36)$$\left[n\right]=\left[\begin{array}{cccccc} n_{x} & 0 & 0 & n_{y} & 0 & n_{z}\\ 0 & d_{y} & 0 & n_{x} & n_{z} & 0\\ 0 & 0 & d_{z} & 0 & n_{y} & n_{x} \end{array}\right]$$

The relationship between stress field and displacement was defined by Clapeyron’s theorem [23]
(37)$$\overset{\;}{\underset{V}{\int }}{\left\{\sigma \right\}}^{T}{\left[\partial \right]}^{T}\left\{u\right\}dV=\overset{\;}{\underset{V}{\int }}{\left\{u\right\}}^{T}\left[n\right]\left\{\sigma \right\}dS-\overset{\;}{\underset{V}{\int }}{\left\{u\right\}}^{T}\left[\partial \right]\left\{\sigma \right\}dV$$

This is a consequence of the Gauss integral theorem and is essentially a comparison of the work of external and internal forces [23].

There are several of these elements found on the structure. The beam one-dimensional element, which is shown in the figure (Figure 6).

This has six degrees of freedom and its displacement vector is as follows
(38)$${\left\{u\right\}}^{T}={\left\{u_{1},v_{1},\varphi_{1},v_{2},u_{2},\varphi_{2}\right\}}^{T}$$

When changing the position of a specific point of an element, this movement affects the points that are connected to it. The influence depends on the stiffness of the element, which is expressed by stiffness constants that depend on the material and cross-sectional characteristics of the element and its boundary conditions; this is the aforementioned material stiffness matrix [*D*] in general form, while also allowing for the non-linear behavior of the material in different directions. Displacement of points on deformed one-dimensional element is presented on Figure 7.
(39)$$\left\{f\right\}=\left[k\right]\left\{u\right\}$$

The deformed element has directly defined border conditions. The element’s material and cross-section parameters are known: *E*—modulus of elasticity, *I*—quadratic moment of section area, *L*—length. It is possible to create a stiffness matrix and evaluate forces and moments accordingly (39) for the element.


(40)
$$\left\{\begin{array}{c} f_{x1}\\ f_{y2}\\ m_{1}\\ f_{x2}\\ f_{y2}\\ m_{2} \end{array}\right\}=\left[\begin{array}{cccccc} k_{11} & k_{12} & k_{13} & k_{14} & k_{15} & k_{16}\\ k_{21} & k_{22} & k_{23} & k_{24} & k_{25} & k_{26}\\ k_{31} & k_{32} & k_{33} & k_{34} & k_{35} & k_{36}\\ k_{41} & k_{42} & k_{43} & k_{44} & k_{45} & k_{46}\\ k_{51} & k_{52} & k_{53} & k_{54} & k_{55} & k_{56}\\ k_{61} & k_{62} & k_{63} & k_{64} & k_{65} & k_{66} \end{array}\right]\left\{\begin{array}{c} u_{1}\\ v_{1}\\ \varphi_{1}\\ u_{2}\\ v_{2}\\ \varphi_{2} \end{array}\right\}$$


Connected elements can be solved the same way. Their resulting forces and moments are defined by a global stiffness matrix and a global deformation vector, which depends on the geometric positions of the elements, their interconnection and the boundary conditions of the system.
(41)$$\left\{F\right\}=\left[K\right]\left\{U\right\}$$

However, when solving such problems in practice, it is not always possible to define the deformation vector in advance; rather, the external influences are defined by the load vector, or forces and moments. Additionally, the real calculated unknown is the deformations of the nodes.
(42)$$\left\{U\right\}={\left[K\right]}^{-1}\left\{F\right\}$$

This problem can be solved analytically, but the disadvantage remains that with the large number of elements used in solving practical problems, it is time consuming and, in most cases, impossible. Therefore, the numerical methods mentioned above are chosen to approximate the result. This is where modern computing comes to help.

The well-known method of solving this numerical model, assuming an approximation to the result, is the variational method, which works with a so-called functional that represents the total energy of the system [24]. If the state is equilibrium, this functional takes on a minimum value under specified boundary conditions, and the process by which it reaches that minimum is called variation in the functional, hence the name. Under specified boundary conditions based on Hooke’s law, the variational model takes the form [24,25].
(43)$$I=\int_{0}^{L}\left(E\left(x\right){\left(\frac{du\left(x\right)}{dx}\right)}^{2}-2u\left(x\right)f\left(x\right)\right)dx+{\left.2\left(-\sigma \left(x\right)u\left(x\right)\right)\right|}_{L}$$

For the solution, it is necessary to find a function *u*(*x*) such that it minimizes the functional, which is written as [26].
(44)$$\delta I\left(x,u\left(x\right),u^{\prime }\left(x\right)\right)=0$$
a symbol *δ* is the first variation of the functional in the variation model [10].

Finite element geometry selection:

For the foam concrete specimens, based on experimental measurements, three-dimensional elements were selected for the finite element method model development. The elements were chosen because of a more accurate response and greater agreement with the experiment itself.

The use of one-dimensional elements is not directly suitable for this particular type of material as it has different mechanical properties in compression and tension and the member element does not accurately represent the behavior that takes place in the cross-section as it does not have a sufficient number of elements in the vertical direction where the variable stresses are located.

However, the use of two-dimensional elements would already be possible, since it is in this plane that the element is loaded, and the most important progression of internal forces and stresses can be captured.

However, three-dimensional elements were used (Figure 8). Mainly, the stakes of the loading method of each foam concrete specimen, which was implemented in the experiment. The load was placed on a part of the structure which could be simulated on a two-dimensional element. However, it was not along the whole width, so it was more convenient to model the structure with three-dimensional elements.

Deformation vector of a three-dimensional element:(45)$${\left\{u\right\}}^{T}=\left\{u_{1},v_{1},w_{1},u_{2},v_{2},w_{2},u_{3},v_{3},w_{3},u_{4},v_{4},w_{4},u_{5},v_{5},w_{5},u_{6},v_{6},w_{6},u_{7},v_{7},w_{7},u_{8},v_{8},w_{8}\right\}$$

The rotation of the nodes is defined by the spatial rotation of the whole element, therefore there is no component of the rotation of the nodes in the deformation vector.

## 3. Results

### 3.1. Results of Analytic Models

Two computational models were created. Of these, one computational model was based on the theoretical basis of a unilaterally embedded beam, as it was representative of the experiment itself.

Many authors use four-point bending to determine the flexural properties of foam concrete, which detects the failure of the material in pure bending, since no shear force enters the internal forces. However, this is not true in a cantilever test (Figure 9 and Figure 10) and the possibility of the results being influenced by this constant shear force along the cross-section must be considered (Table 2). This type of test was chosen to investigate the possibility of delamination of the composite material of foam concrete and cement concrete. However, it turned out that the foam concrete itself has very complicated properties and needs to be investigated in more detail.

The second computational model, from which the material model or stress–strain diagram of the foam concrete was later defined in an iterative manner, is based on numerical finite element methods. Specifically, the original Ansys numerical system with its original subsystems was used in combination with the modern Ansys workbench. The model was based on three-dimensional elements, mainly because of the way the specimens were loaded to simulate the experiment more accurately.

### 3.2. Results of Experimental Measurements

In the experimental investigation, the special numerical signs for all tested specimens were created. In this numerical identification sign is implemented the date of specimen manufacture unit mass after 28 days and the type of series. All presented specimens’ results were selected as representative examples from the median evaluated of the series.

#### Calibration of Experimental Setup

The experimental setup was calibrated using a calibration steel rod and tested using multiple cement concrete specimens, which were loaded till collapse and then evaluated (Figure 11). The results were comparable with standardized classes of concrete and therefore the setup was considered as calibrated correctly (Figure 12).

### 3.3. Results of Experimental Measurements of Foam Concrete

For the presentation of the results, several representative specimens were selected and further used for the evaluation of the material model of foam concrete in software based on the finite element method. These specimens were also used for comparison with the predicted linear mechanical properties of foam concrete in the following chapters. Other specimens were excluded from the work due to incorrect fitting in the measuring line and thus their failure mode or measured nonrepresentative values. All tested series were performed and evaluated until collapse (Figure 13, Figure 14 and Figure 15).

From the evaluated specimen data, it is possible to observe a nonlinear deformation progression depending on the specimen loading, which is observed in many studies of international as well as Slovak authors. Since the work is also concerned with the development of a material model of foam concrete, this particular specimen is chosen together with its deformation curve. The material model will represent an increase in deformation similar to this particular curve. This is justified by the use of FEA software and its ability to simulate a non-linear material model, which is represented by specifying specific values for the stress–strain diagram. However, the values that the FEM model will output may be influenced by the actual behavior of the mathematical model, which will overlay the curve for the model to work properly.

It is a fact that at loads ranging from 90 N to 120 N, which is approximately 0.14 MPa to 0.19 MPa for this type of specimen, a large number of specimens slip. It is quite probable that this may represent the aforementioned pore shrinkage observed by the authors of the articles in the journal *Materials and Design*, under the title “Experimental and numerical investigation of influence of air-voids on the compressive behavior of foamed concrete”. However, this statement needs to be further verified by experiments that will focus directly on this type of deformation. It should also be pointed out that this is an element subjected to bending and not to compression, as is the case in the literature referred to above. Thus, it is very likely that the specimens or the measuring device may be subject to slip deformation.

The deformation, deflection and load force data were recorded from the individual specimens. Measurement errors were systematically removed from these data along with the specimens that were determined to be non-conforming for the evaluation of the experiment. From the suitable specimens, these data were entered into a table and their flexural strength was evaluated; however, it should be noted that this is not a pure bending test, but a bending accompanied by shear, since it is a cantilever bending test. The elastic modulus of the specimens themselves was also evaluated, which was determined mainly from their linearly behaving part. The average of these values was also evaluated to unify the results. It should be noted that these are experimentally measured values. Below, Table 3 shows the first six specimens which are the specimens that show the average and normal values of the flexural strength and modulus of elasticity, respectively.

### 3.4. Results of Finite Element Method

The analytical model that was used to evaluate the specimens varies depending on the specimen parameters. Each specimen has its own dimensional parameters from which its quadratic moment of cross-section area is derived as follows:(46)$$I=\int_{A}^{\;}y^{2}dA$$
this formula is used in relation to (4), determining the bending moment on the specimen. Thus, it can be said that this is a fully linear calculation where the Navier Bernoulli hypothesis and Hooke’s law are considered. In the case of the analytical model, an ideal specimen of cross-section dimensions 100 mm × 100 mm was used to represent the form used in the experiment. Of course, the fabricated specimens themselves had minor dimensional imperfections, which are noted in the table. However, these are not flaws that could extremely affect the final behavior of the specimen and of course this is taken into account when evaluating the experiments themselves.

Another parameter in the analytical calculation model is the modulus of elasticity of foam concrete. In this case, the modulus of elasticity *E* mentioned in Section 3.2 is not considered, since it is the compressive or tensile modulus, and this particular case is solved in bending. Therefore, the average measured modulus of elasticity of the selected specimens *E* = 520 MPa is used for comparison.

Additionally, of course, the boundary conditions of the cantilever model, such as load and support at the point of insertion are correctly modeled.

The result is a linear deflection in the direction of the “*y*” axis, denoted by “*w*” in millimeters, and an increasing force “*F*” in Newtons. The progression of these values will be indicated by the green dashed line in captures 3.4.2 and 3.4.3.

#### 3.4.1. Definition of Stress Strain Diagram of Foam Concrete FC500

To compare the average values, a material model of foam concrete was constructed iteratively in Ansys Mechanical APDL 2022 R1 using tensile and compressive parameters.

The tensile parameters for the model were taken from the work of Matej Prišč [15], who performed tensile tests of foam concrete on a similar measuring line adapted for tensile tests. The tests were made on experimental specimens and the measurement results were varied. It was necessary to summarize these data and to derive an average stress–strain of the foam concrete in tension. The diagram was then partially modified to correlate the results with the comparison specimen of the cantilever test experiment.

Using a number of scientific articles, a number of diagrams of the material under pressure have been compiled [17,27,28,29]. These diagrams were mostly made during material in compression tests.

The material characteristics thus created were entered into Ansys via a non-linear material model called “Cast-Iron” (Figure 16), which acts linearly in the first part and then starts to behave non-linearly according to the specified material characteristics. This material model is used specifically for compressible materials, which are brittle and have an order of magnitude lower tensile than compressive load capacity. It is based on Coulomb-Mohr theory, which divides the stress acting on a cross-section into four quadrants depending on the stress in compression or tension. This function is enclosed by the maximum stress values that the cross-section can withstand, whereby the *x*-axis shows the axial stresses in the *x*-axis direction, either tensile or compressive, and the *y*-axis shows the axial compressive strengths in the *y*-axis direction. Thus, at a forty-five-degree angle in the first quadrant and in the third quadrant, we are talking about biaxial tensile and compressive stresses, respectively, and in the second and fourth quadrants, the model represents tensile stresses in one axis and compressive stresses in the other axis.

However, the input values implemented to the material model did not accurately represent the behavior of the specimen. The tensile stress–strain diagram was relatively unchanged, except for minor modifications. It should be noted, however, that the original stress–strain dependence in compression prior to the iterative changes was made from specimens that were loaded in pure compression. Already, as Coulomb-Mohr theory shows, the material has a higher resistance in compression than in tension. A cantilever test experiment was realized in which tensile failure of the specimen is assumed. Thus, it was necessary to refine the details of the stress–strain in compression to tensile failure. The stress–strains taken from different authors were inaccurate or considered as linear in such low stresses.

Since it is still a non-linear material, it is not possible to assume a linear distribution of stresses across the cross section, which was observed when the stress–strain was iterated (Figure 17). This is meaningful when considering the shape of the stress–strain curve. Deformations from increasing stresses are more evident in the compression section than in the tension section, depending on the resulting tensile and compressive parameters of the stress–strain diagram.

This behavior caused the cross-sectional failure due to compression in the case where the working diagram of the foam concrete in compression was completed at the same stress as in tension. In order to increase the load, it was necessary to add values for the resistance of the material in compression above the stress of 0.3 MPa, at which collapse is considered in the case of tensile loading. This simulates the aforementioned non-linear stress distribution across the cross section. It should also be noted that stresses above 0.3 MPa in compression should not be considered as correct, since this is a region that is defined when the specimen collapses in tension and cannot be assumed to be consistent.

It should be noted that this graph (Figure 18) is experimental, and cannot be considered applicable in practice, as it is based on a relatively small number of specimens tested. The material has been tested under known conditions, mainly at known moisture content and bulk density, which can vary considerably due to environmental and manufacturing influences. The method of preparation may also vary, which affects the amount, size and distribution of pores in the structure.

#### 3.4.2. Comparison of Theoretical Approaches FEM and Analytic

The experimental stress–strain diagram of the foam concrete FC500 was created and iterated, an algorithm was developed to compose a finite element numerical model that was loaded in a very similar to identical manner to the experiment itself. The deflections of the modelled specimen were measured at the same point as in the experiment and the analytical model, respectively. Subsequently, the numerical model produced in Ansys Mechanical APDL (original Ansys) was compared with the predicted analytical model of foam concrete, using the linear assumptions of the theories already mentioned.

It can be observed from Figure 19 that the linear values of the analytical model based on double integration and linear material characteristics are almost identical to the numerical model developed by FEM using the established experimental working diagram. However, this changes after reaching approximately 120 N, which represents a stress of approximately 0.19 MPa. Similar stresses were observed for several specimen slips that could not be included in the results. At this stress, due to the nature of the material model, the strains start to increase more significantly than in the analytical model. The ultimate load capacity can be fixed for these mathematical models to an indeterminate, it is purely dependent on the chosen parameters. However, in the numerical model that has been used it is necessary to produce a working diagram that would not allow the complete collapse of the specimen after exceeding approximately 0.35 MPa in compression and 0.3 MPa in tension as specified. These values are derived from the iterative procedure by which the stress–strain diagram itself was created. This behavior of the material can also be observed in experimental measurements. The other stress–strain relationship (indicated by the dashed line in the stress–strain diagram) cannot be determined from this experiment. This is due to the fact that there is a collapse of the specimen due to tensile stress.

#### 3.4.3. Experimental Measurements Compared with the Analytical Models

For the comparison of experimental measurements and analytical models, two sets of data were selected, which are divided into two parts, namely those suitable for the determination of material properties, and data that are not suitable for further use as a demonstration.

The aforementioned double integration method was used to demonstrate the comparison between experimental measurements and analytical models on individual specimens. It is clear from the type of stress on the specimens that boundary conditions would be determined to establish the relevance of the results. The upper boundary condition is prompted by a compressive modulus of elasticity of 1.2 MPa, while the lower boundary is bounded by a tensile modulus of 0.3 MPa, which is based on the experimental data of the cited papers mentioned in chapter three. It is evident that most of the specimens selected as presentable fall within the zone bounded by these conditions. The third linear model used in the graph (Figure 20) under the name tension is a linear model with an applied elastic modulus of 0.35 MPa, which is taken from the literature mentioned in chapter three.

From these results, it is noticeable that the experimental data behave in a non-linear manner, and it is not possible to accurately capture their behavior in a real situation. Therefore, under this assumption, material models were further constructed using numerical methods.

The next graph (Figure 21) depicts the non-conforming specimens in terms of their progress as a function of the specified boundary conditions. The assumption is that the specimens have been incorrectly made or have slipped due to deposition. It is also possible that they have been damaged by over-tightening of the anchor blocks or an increase in deformation due to line stiffness and instability.

#### 3.4.4. Experimental Measurements Compared with the FEM Models

Since the material model was based on the assumption that there is pore shearing or weakening of the material in compression by increasing stress, the deflection behavior of the numerical model is highly nonlinear. The example of normal stress σ_1_ redistribution on the central plane is presented in Figure 22 as one force step.

The stresses observed in the FEM model represent the non-linear behavior of the material where plasticization occurs in the tensile area. This behavior leads to an increase in stresses at the lower compressive area above the theoretical linear limit considered and, hence, it is necessary to use a stress–strain diagram that achieves bearing stresses up to 0.47 MPa; however, above this limit, the stress–strain diagram cannot be iterated in this case. This is represented by the dashed line in the diagram (Figure 23).

Since this specimen (a) from Figure 7 used as an iteration guide was used in the iteration of the experimental stress–strain diagram of foam concrete, its deformation behavior is almost identical to the proposed material model. It well corelates with the linear part of the behavior of the specimen but also corelates with the progression of the onset of nonlinearity and increases with it until the phase where the cross section is partially linear again. In the collapse phase of the specimen, the material model continues to be linear, because of the functionality of numerical models. It should be noted that this is thus an idealized state of the diagram applied to a particular specimen, and the material may behave in other ways according to other results, which in this particular stress state can in some cases be characterized by a linear dependance.

## 4. Discussion

Generally, this type of material can be used in road construction, but it can also find its application in building construction. It should be noted, however, that we know of materials which have a far higher tensile strength than cement concrete, and it is possible to use these together in combination with cement concrete.

Since cement concrete has better tensile properties than foam concrete, it is possible to load the overall cross-section more, since without the added composite the foam concrete itself collapses due to tensile overload. The following several observed facts can be discussed in further research:

Several specimens collapsed without delamination (Figure 24).

Since the clamps in some of the observed experiments were designed without surface correction, delamination of the two layers due to shear, which is present in this type of bending test, were observed (Figure 25, Figure 26, Figure 27 and Figure 28).

When the cement concrete cross-section is overloaded in tension in the upper surface, the element plasticizes and redistributes tensile stresses into the foam concrete layer below, which subsequently collapses. Shear delamination is probably observed not only at the point of failure but throughout the cross-section. This fact needs to be investigated further.

It can be clearly seen that the cement concrete is subjected to much greater tensile stresses than when the cross-section was purely composed of foam concrete (Figure 29, Figure 30 and Figure 31). This is due to its greater tensile capacity. It therefore appears stiffer and can withstand a greater amount of load. The element thus combined could also be loaded with a greater amount of force.

It shows the evolution of different stresses across the cross-section at the point of contact of two different materials with different material characteristics. The magnitude of the stresses varies with increasing force. In this case, it is the condition at a load of 300 N, as in the case of a no composite system.

Considering a 5% tensile strength fraction of cement concrete of class C20/25 *f_ctk_*_,0.05_ = 1.5 MPa, it is possible to load a cross-section with a force value of 250 N in the simulation, considering 50 mm of foam concrete and 20 mm of cement concrete rigidly connected. The cement concrete in tension reaches stresses of up to 1.1 MPa, while the foam concrete only takes part of the stress.

The FC500 polymer foam concrete material is investigated by authors in Slovak institutions by researchers at the University of Zilina [30,31,32]. There are several methods of experimental utilization in transport structures but only in regard to measurements and observations [33,34]. For further research, many parametric analyses must be realized before this material (FC500) can be used as a standard for load bearing structures (rheology, fatigue, frost-defrost processes, etc.).

## Figures and Tables

**Figure 1 polymers-14-03786-f001:**
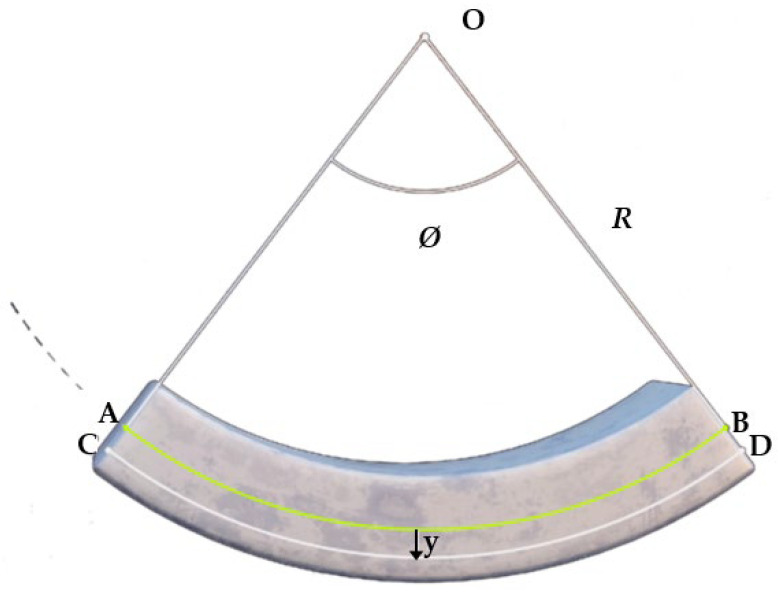
Beam loaded by a pair of moments to derive the double integration method.

**Figure 2 polymers-14-03786-f002:**
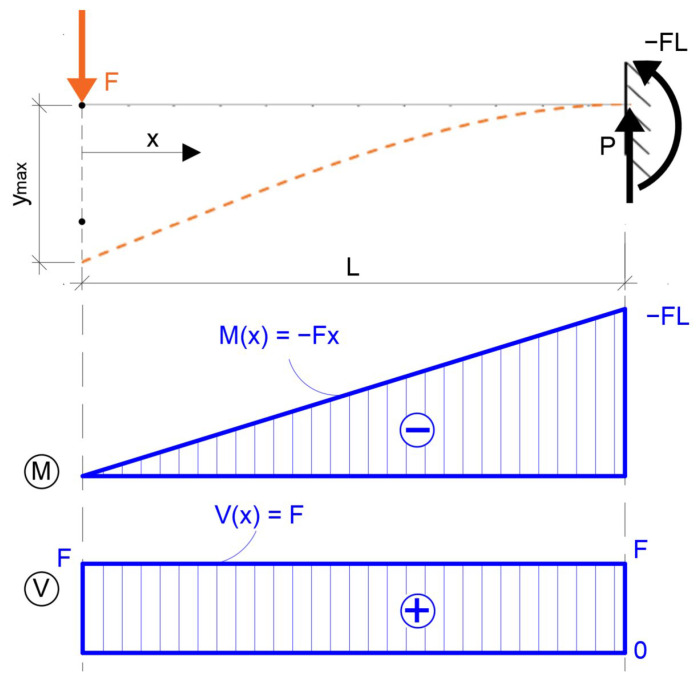
Cantilever beam and its internal forces from single concentrated force.

**Figure 3 polymers-14-03786-f003:**
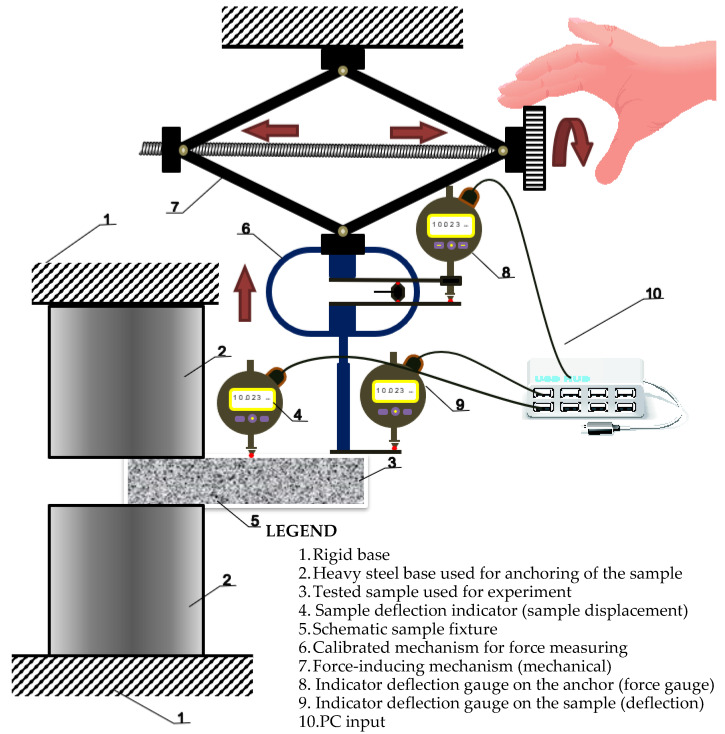
Scheme of experimental measuring setup.

**Figure 4 polymers-14-03786-f004:**
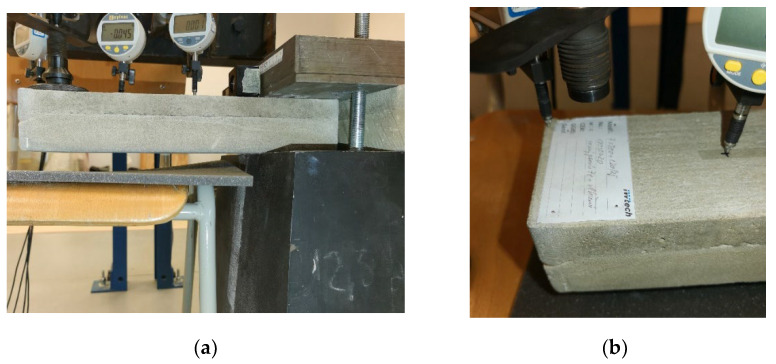
Detail of (**a**) experimental measuring setup; (**b**) and labeling of specimens.

**Figure 5 polymers-14-03786-f005:**
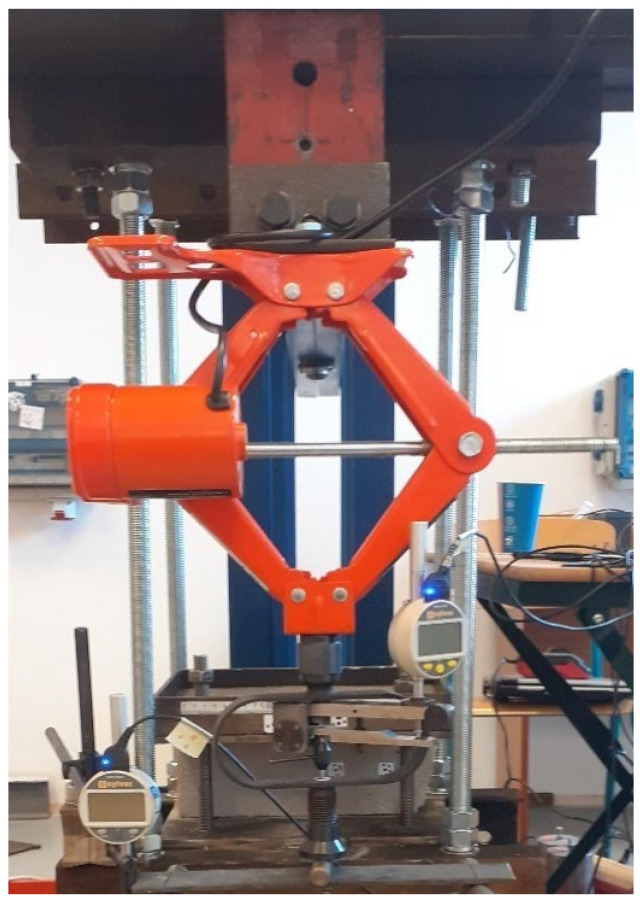
Mechanism used to introduce the load into the specimen.

**Figure 6 polymers-14-03786-f006:**
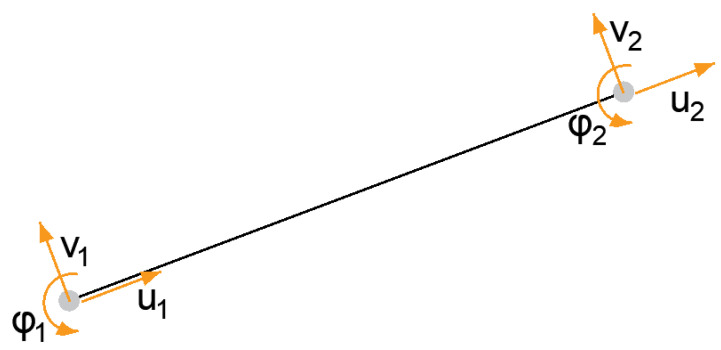
Schematic of a one-dimensional element.

**Figure 7 polymers-14-03786-f007:**
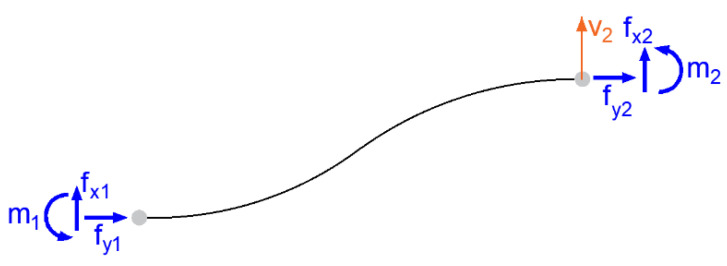
Displacement of point 2 in horizontal direction.

**Figure 8 polymers-14-03786-f008:**
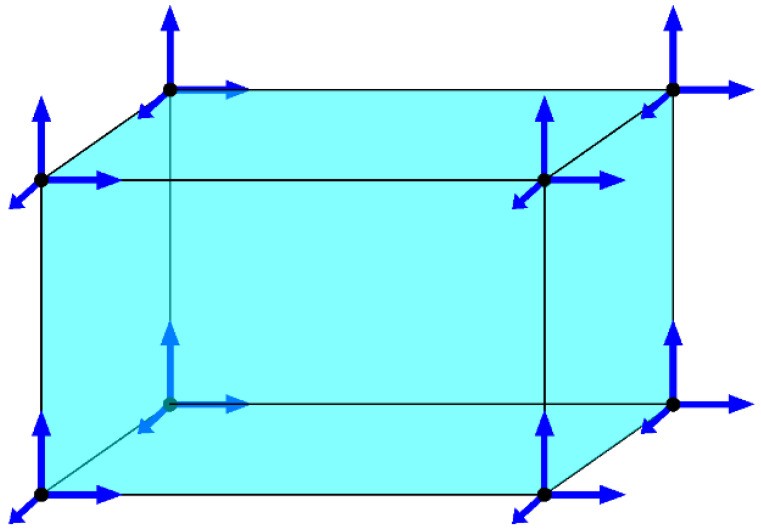
Shape of the three-dimensional element used in the FEM calculation.

**Figure 9 polymers-14-03786-f009:**
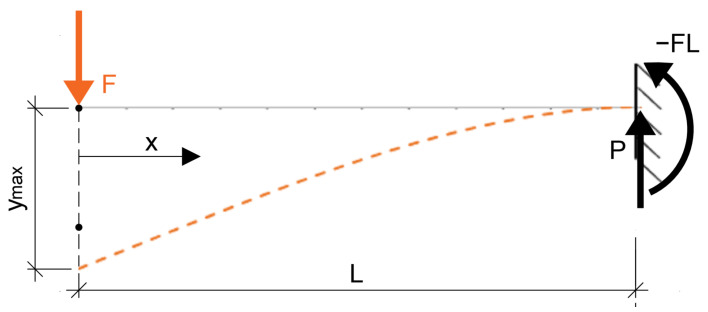
Schematic of the analytical load test used.

**Figure 10 polymers-14-03786-f010:**
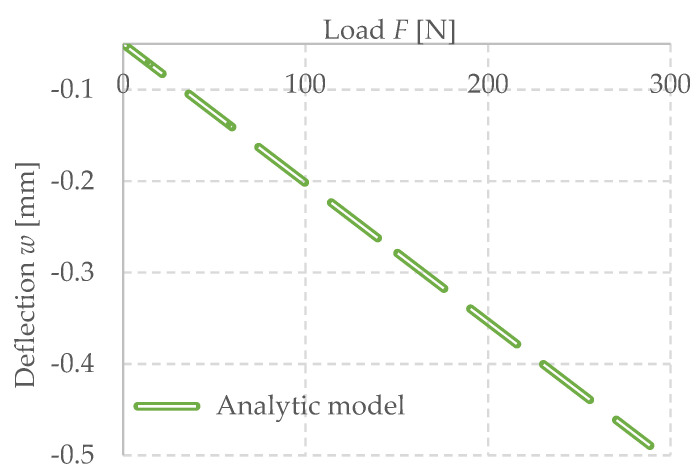
Schematic dependance of the analytical load test used.

**Figure 11 polymers-14-03786-f011:**
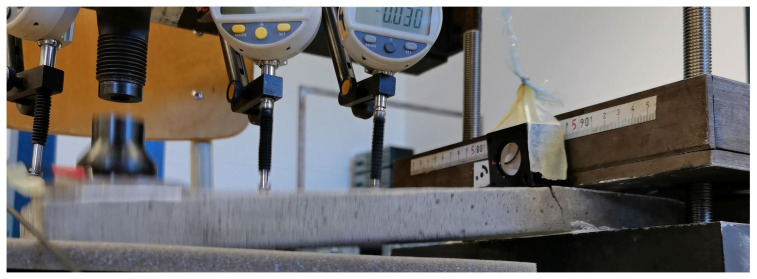
Collapse of the CC specimen 5200420.

**Figure 12 polymers-14-03786-f012:**
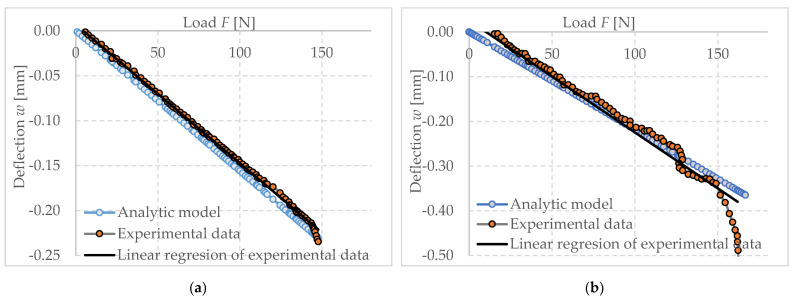
Dependence of deflection (cantilever end) on force (load of the cantilever end) investigated CC specimen 5200420, (**a**) with used calibration of measuring system (**b**) not corrected data.

**Figure 13 polymers-14-03786-f013:**
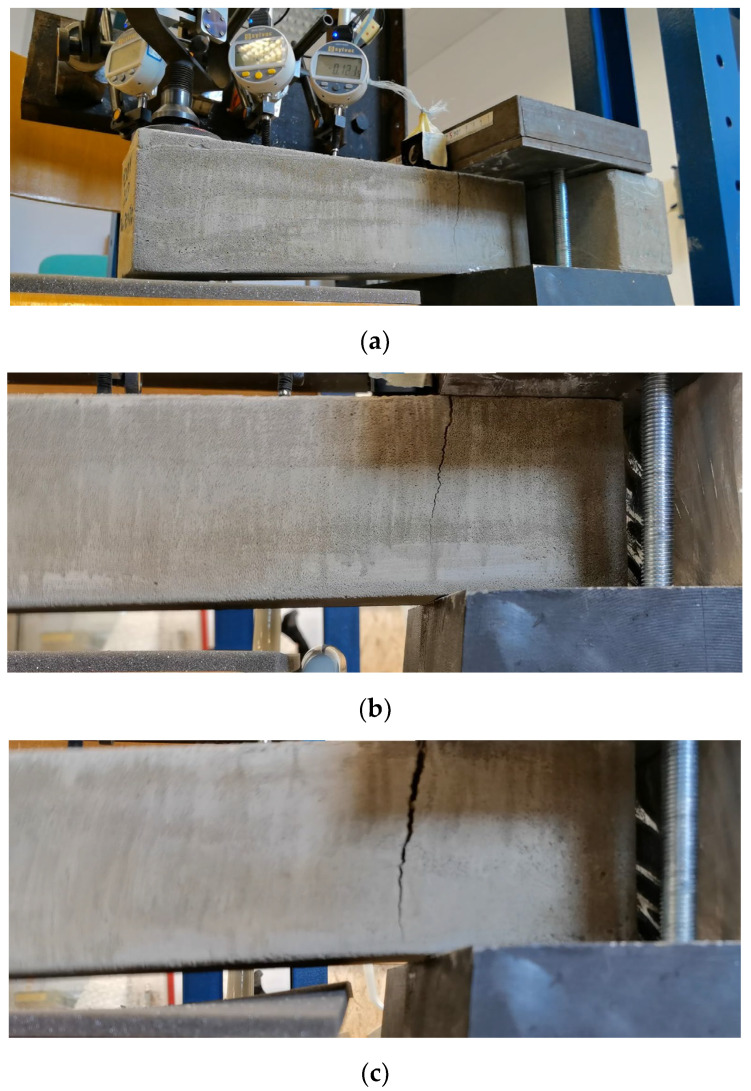
Crack development FC specimen 35030420: (**a**) collapse—first stage, creation of the crack; (**b**) second stage—crack propagation over specimen cross-section; (**c**) third stage—total collapse, crack over full cross-section.

**Figure 14 polymers-14-03786-f014:**
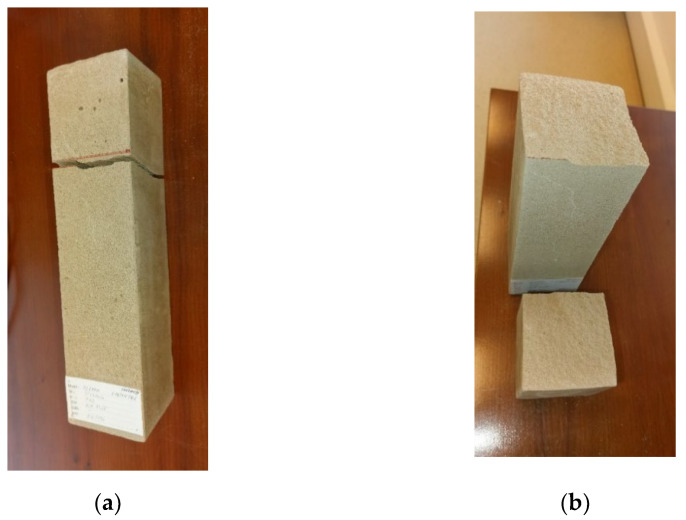
FC specimen 35030420–unclamped: (**a**) cross-section view of the cracked “lines” on the sides of the block; (**b**) surface shape of the cracked area.

**Figure 15 polymers-14-03786-f015:**
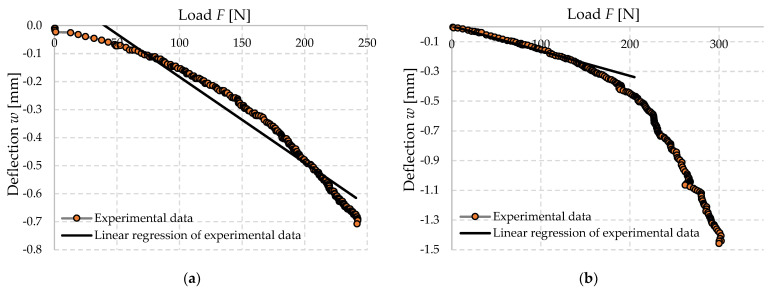
Dependence of deflection (cantilever end) on force (load of the cantilever end) investigated FC500 specimen 35030420, (**a**) not corrected data, (**b**) with used calibration of measuring system.

**Figure 16 polymers-14-03786-f016:**
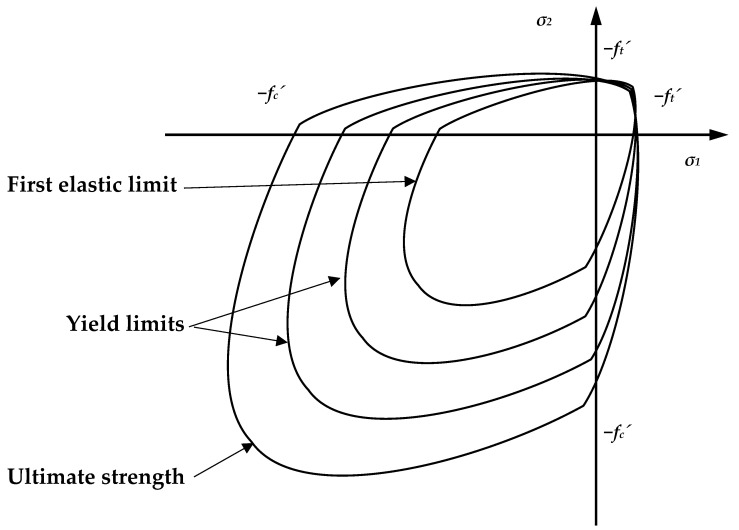
Composite material model principle “Cast− Iron”.

**Figure 17 polymers-14-03786-f017:**
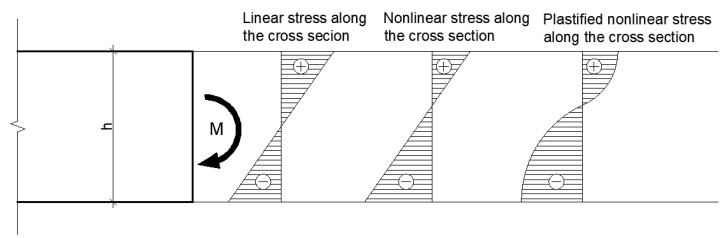
Stress redistribution over cross-section in box specimen model–comparison between linear and no-linear stage.

**Figure 18 polymers-14-03786-f018:**
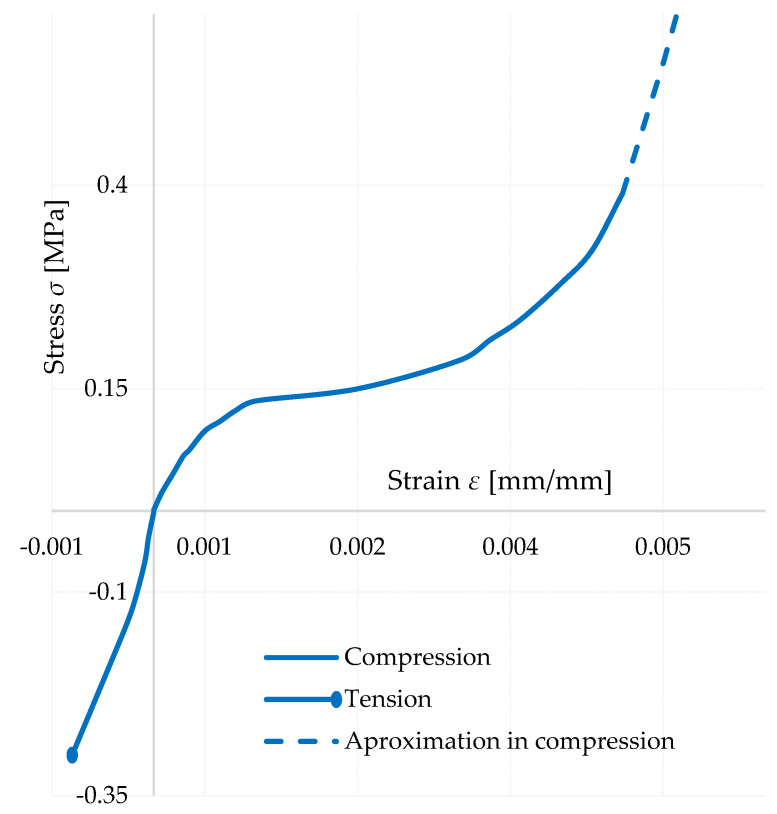
Stress–strain diagram–FC500 used for evaluation.

**Figure 19 polymers-14-03786-f019:**
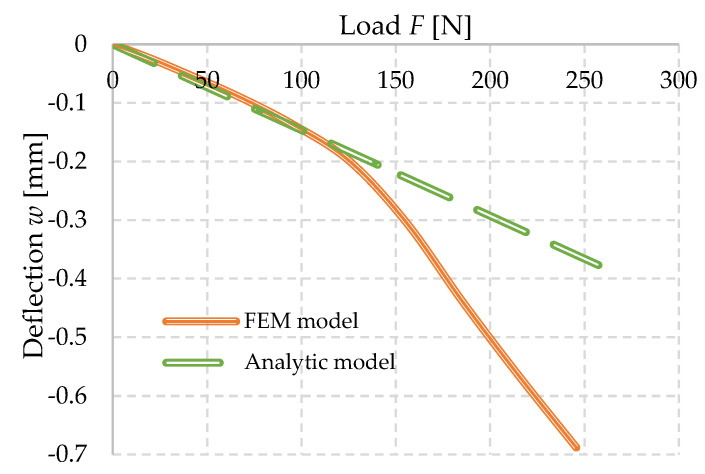
Comparison of theoretical results–dependence of deflection (cantilever end) on force (load of the cantilever end).

**Figure 20 polymers-14-03786-f020:**
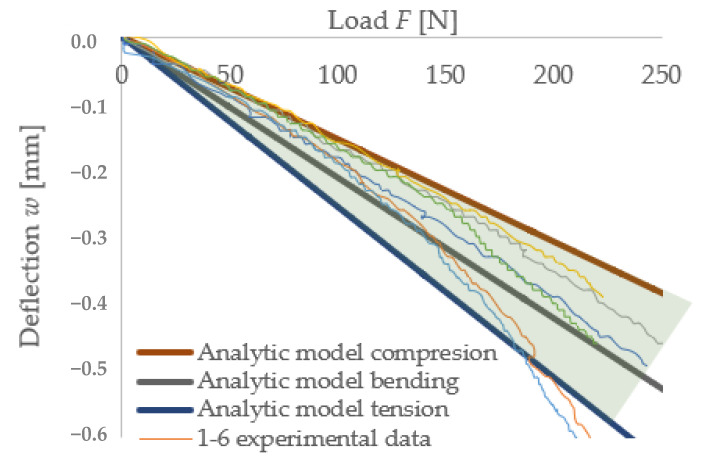
Comparison of experimental and analytical results.

**Figure 21 polymers-14-03786-f021:**
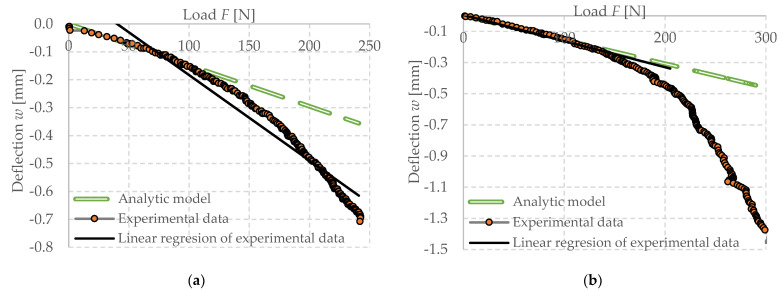
Comparison of analytical dependence of deflection (cantilever end) on force (load of the cantilever end)–investigated material FC500 (**a**) specimen 34030420 medium load capacity; (**b**) specimen 32030420 highest load capacity.

**Figure 22 polymers-14-03786-f022:**
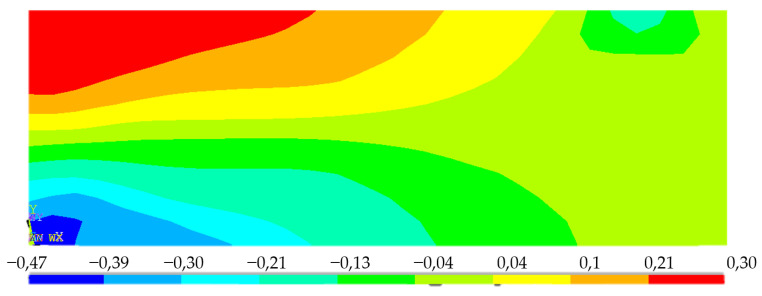
Example of FEM results–main normal stress *σ*_1_ redistribution–longitudinal-section, load *F* = 280 N.

**Figure 23 polymers-14-03786-f023:**
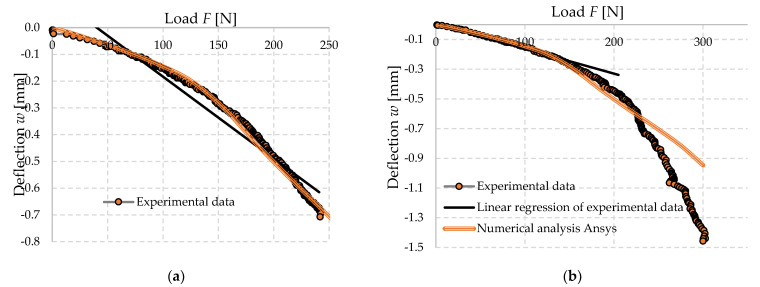
Comparison of numerical dependence of deflection (cantilever end) on force (load of the cantilever end)–investigated material FC500 (**a**) specimen 34030420 medium load capacity; (**b**) specimen 32030420 highest load capacity.

**Figure 24 polymers-14-03786-f024:**
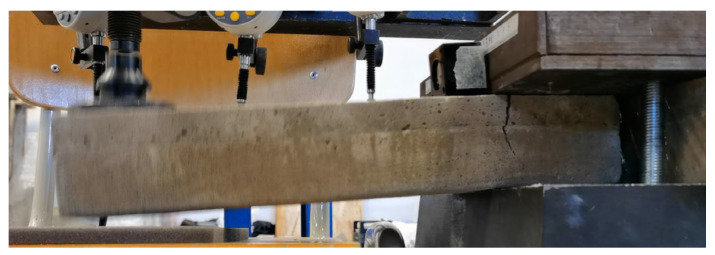
Collapse of specimen 19030420 after load test (composite FC and CC) without delamination.

**Figure 25 polymers-14-03786-f025:**
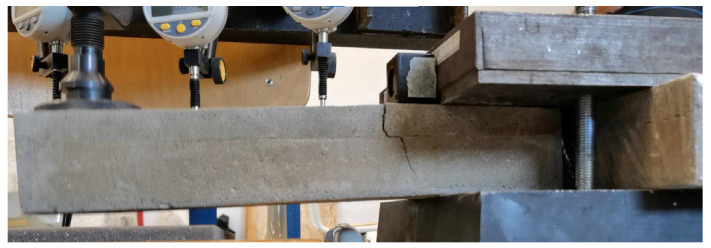
Delamination of specimen 22030420 after load test (composite FC and CC).

**Figure 26 polymers-14-03786-f026:**
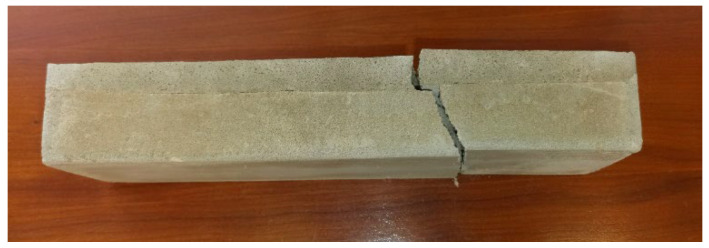
Detail of delamination-specimen 22030420.

**Figure 27 polymers-14-03786-f027:**
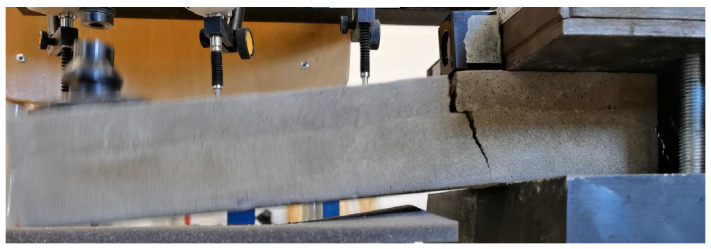
Delamination of specimen 24030420 after load test (composite FC and CC).

**Figure 28 polymers-14-03786-f028:**
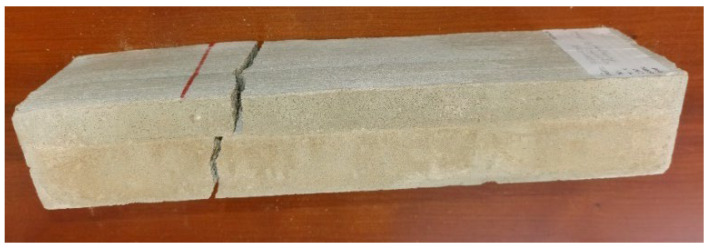
Detail of delamination-specimen 24030420.

**Figure 29 polymers-14-03786-f029:**
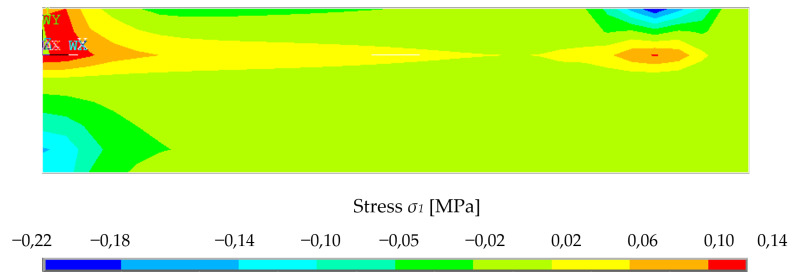
Stresses iso-areas over longitudinal-section for interaction of two materials.

**Figure 30 polymers-14-03786-f030:**
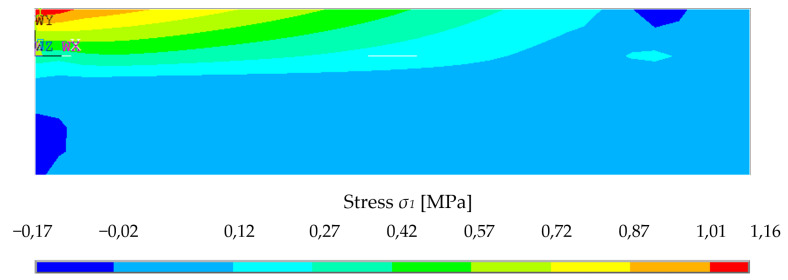
Stresses iso-areas over longitudinal-section.

**Figure 31 polymers-14-03786-f031:**
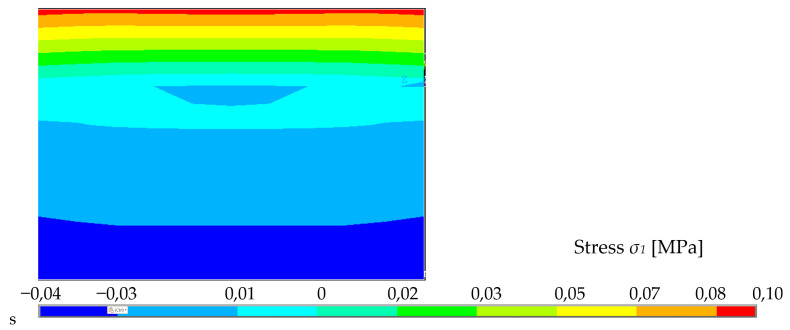
Stresses iso-areas over cross-section close to clamping.

**Table 1 polymers-14-03786-t001:** Selected material characteristics of foam concrete.

Type	*E* [GPa]		*v*	*ρ*	*f_cf_*	*f_c_*	*f_t_*
	Tension	Compression	[-]	[kg/m^3^]	[MPa]	[MPa]	[MPa]
FC 500	-	1.2~2.5	0.11	584	0.35	0.472	-
	-	1.2~2.5	0.2	584	0.35	0.708	-
	0.34	-	-	-	-	-	0.1
	-	-	-	512	0.36	-	-
Not specified	-	1.2	0.2	1600	1.86	-	-
	0.56	-	-	650	-	1.9	0.28
	-	2.6	-	1000	-	2.6	0.82
	0.24	-	-	-	-	7.74	-
	-	-	-	400	-	1.16	0.1
	-	-	-	500	-	2	0.2
	-	-	-	600	-	3.5	0.3
	-	-	-	500	-	2.8	-

**Table 2 polymers-14-03786-t002:** Selected material characteristics of foam concrete [1,2,3,4,5,6].

Type	*E* [GPa]		*v*	*ρ*	*f_cf_*	*f_c_*	*f_t_*
	Tension	Compression	[-]	[kg/m^3^]	[MPa]	[MPa]	[MPa]
FC500	0.3	1.2	0.2	500	0.35	1	0.15

**Table 3 polymers-14-03786-t003:** Selected material properties of the FC [1,2,3,4,5,6].

Specimen	*b* [mm]	*h* [mm]	*m* [g]	*ρ* [kg/m^3^]	*F_fat_* [N]	*w_fat_* [mm]	*σ_fat_* [MPa]	*E_exp_* [MPa]
31030420	100.5	104.2	2512	790	288.7	−0.48	0.41	495
32030420	97.4	103.6	2437	800	297.5	−1.36	0.44	495
33030420	101.8	104.9	2559	790	261.9	−0.38	0.36	475
34030420	102.4	105.1	2531	780	222.8	−0.31	0.30	575
35030420	97.1	105.1	2435	790	240.9	−0.67	0.35	530
36030420	100.4	105.1	2518	790	220.5	−0.38	0.31	550

## Data Availability

The data presented in this study are available on request from the corresponding author. At the time the project was carried out, there was no obligation to make the data publicly available.

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
