# Peer review of "Polymer Foam Concrete FC500 Material Behavior and Its Interaction in a Composite Structure with Standard Cement Concrete Using Small Scale Tests"

_polymers, 2022, doi:10.3390/polym14183786_

Round 1

Reviewer 1 Report

In this manuscript, the authors investigated the material properties of FC500 foam concrete. The methods chosen for the investigation of the material properties were: small-scale cantilever test, standard tensile, and compression test. The cantilever test was carried out both for the individual components of the FC500 composite and the cement concrete but also as a compact composite. Numerical simulation models were developed to correlate the individual results in order to verify the results of the uniaxial tests. The cement concrete is subjected to much greater tensile stresses than when the cross-section was purely composed of foam concrete which is due to its greater tensile capacity. It, therefore, appears stiffer and can withstand a greater amount of load. The element thus combined could also be loaded with a greater amount of force. This study is good and important for the preparation of foam concrete as well as its method of preparation. Compared to ordinary concrete, foam concrete has many advantages. The interpretations and discussion are well justified by the results. In addition, the quantity and quality of the figures are appropriate. We believe that this research subject is promising for developing foam concrete with high mechanical.

Summary: I recommend publishing this manuscript after considering my comments on the attached file.

Author Response

Many thanks for the review, the responses are in the attached file.

Reviewer 2 Report

This paper mainly studies the material properties of FC500 foam concrete. The cantilever test was carried out both for the individual components of the FC500 composite and the cement concrete but also as a compact composite. In order to verify the results of uniaxial tests, a numerical simulation model was developed to correlate the results. The research done in the article is not practical, the theoretical analysis is shallow, the description of the article is not clear, there are many format problems in the article, the degree of refinement is not high, and it is recommended to reject the manuscript.

The specific opinions are as follows

1. The abstract is too concise. It is suggested to describe it in the order of innovation, main research contents and important research conclusions.

2. The introduction is not enough, and needs to supplement the research status at home and abroad.

3. Some sentences in this paper lack punctuation marks or punctuation marks are wrong, for example: Cast Iron, it is suggested that the author check the punctuation of the full text; The serial number of the pictures in this article is wrong, and the author is suggested to check it; The pictures in this article are not beautiful enough, for example, the format of the pictures is not uniform, some words in the pictures are blocked, and the font in the pictures is too small. The author is suggested to make relevant adjustments; The format of reference numbers is suggested to be uniform for authors, such as [5,10] or [5] [10].

4. The introduction of finite element and its basic principle in Section 2.4 is too long, so it is recommended to simplify it.

5. The type of foam concrete in this paper is "FC500" or "500FC",  it is suggested that the authors should be numbered uniformly.

6. The number of specimens in the article is too confusing, which makes it difficult for readers to read. The author should clarify the similarities and differences between the specimens "31030420" in Table 3 and the  specimens "32030420, 34030420, 22030420, etc." mentioned by the author in the article, and what kind of test or analysis they are used for. Moreover, there are several errors in the number of specimens in the article.

7. In the analysis in Section 3.2, figures 13-14 are the failure modes of specimen 3200420, but the deflection and load analysis curves in Figure 15 are those of  specimen 35030420, and the analysis in the article is inconsistent.

8. The format of references is incorrect. For example, the titles of references in references 10 and 14 only need to capitalize the first letter of each word.

Author Response

(The authors gave the same response as above.)

Round 2

Reviewer 2 Report

This paper mainly studies the material properties of FC500 foam concrete. The cantilever test was carried out both for the individual components of the FC500 composite and the cement concrete but also as a compact composite. In order to verify the results of uniaxial tests, a numerical simulation model was developed to correlate the results. The article has made detailed amendments to the previous opinions, and the degree of refinement has been improved. It is recommended to accept it.